# In Vitro and In Vivo Cell-Interactions with Electrospun Poly (Lactic-Co-Glycolic Acid) (PLGA): Morphological and Immune Response Analysis

**DOI:** 10.3390/polym14204460

**Published:** 2022-10-21

**Authors:** Ana Chor, Christina Maeda Takiya, Marcos Lopes Dias, Raquel Pires Gonçalves, Tatiana Petithory, Jefferson Cypriano, Leonardo Rodrigues de Andrade, Marcos Farina, Karine Anselme

**Affiliations:** 1Biomineralization Laboratory, Institute of Biomedical Sciences, Federal University of Rio de Janeiro, Rio de Janeiro 21941-902, Brazil; 2Immunopathology Laboratory, Institute of Biophysics Carlos Chagas Filho, Federal University of Rio de Janeiro, Rio de Janeiro 20941-902, Brazil; 3Catalysis Laboratory for Polymerization, Recycling and Biodegradable Polymers, Institute of Macromolecules Professor Eloisa Mano, Federal University of Rio de Janeiro, Rio de Janeiro 21941-598, Brazil; 4Biomaterials—Biointerfaces Laboratory, Institut de Science des Materiaux de Mulhouse, University of Haute-Alsace, CNRS, IS2M UMR 7361, F-68100 Mulhouse, France; 5Padrón-Lins Multi-User Microscopy Unit, Institute of Microbiology Paulo Góes, Federal University of Rio de Janeiro, Rio de Janeiro 21941-902, Brazil

**Keywords:** PLGA, electrospinning, morphology, immune response, microscopy

## Abstract

Random electrospun three-dimensional fiber membranes mimic the extracellular matrix and the interfibrillar spaces promotes the flow of nutrients for cells. Electrospun PLGA membranes were analyzed in vitro and in vivo after being sterilized with gamma radiation and bioactivated with fibronectin or collagen. Madin-Darby Canine Kidney (MDCK) epithelial cells and primary fibroblast-like cells from hamster’s cheek paunch proliferated over time on these membranes, evidencing their good biocompatibility. Cell-free irradiated PLGA membranes implanted on the back of hamsters resulted in a chronic granulomatous inflammatory response, observed after 7, 15, 30 and 90 days. Morphological analysis of implanted PLGA using light microscopy revealed epithelioid cells, Langhans type of multinucleate giant cells (LCs) and multinucleated giant cells (MNGCs) with internalized biomaterial. Lymphocytes increased along time due to undegraded polymer fragments, inducing the accumulation of cells of the phagocytic lineage, and decreased after 90 days post implantation. Myeloperoxidase**^+^** cells increased after 15 days and decreased after 90 days. LCs, MNGCs and capillaries decreased after 90 days. Analysis of implanted PLGA after 7, 15, 30 and 90 days using transmission electron microscope (TEM) showed cells exhibiting internalized PLGA fragments and filopodia surrounding PLGA fragments. Over time, TEM analysis showed less PLGA fragments surrounded by cells without fibrous tissue formation. Accordingly, MNGC constituted a granulomatous reaction around the polymer, which resolves with time, probably preventing a fibrous capsule formation. Finally, this study confirms the biocompatibility of electrospun PLGA membranes and their potential to accelerate the healing process of oral ulcerations in hamsters’ model in association with autologous cells.

## 1. Introduction

In the history of tissue engineering, degradable materials such as poly (lactic acid—PLA) and poly (glycolic acid—PGA) or their combination poly (lactide-co-glycolide—PLGA), being of smooth or rough surfaces, were capable of inducing tissue or organ regeneration [1]. Recently, degradable and non-degradable FDA-approved polymers [2] with bioactivated surfaces were shown boosting tissue engineering and regenerative medicine applications [3,4]. To this end, medical device design, medical device implantation and immune responses play crucial roles, in combination, to modulate immune responses for tissue regeneration [5,6].

Concerning degradation products in the organism, PLA and PGA by-products have been considered to be nontoxic. After degradation, by-products of lactic acid and glycolic acid are similar to endogenous metabolites. Lactic and glycolic acid enter the tri-carboxylic acid cycle and are eliminated as carbon dioxide and water [7] through respiration, feces and urine [8].

Electrospinning is considered a smart and easy technique to produce 3D synthetic devices with a porous and fibrous architecture similar to biological extracellular matrix [9,10]. Particularly, this technique has been reported to be effective because it allows the mixture of different monomers to improve the degradation time for drug or cell delivery to a specific microenvironment [11]. Such devices are at the forefront in the regenerative medicine area improving cell behavior in a specific milieu. Specifically speaking, translational application of electrospun fibers for oral disease treatment has been used including a variety of oral clinical applications [12]. Regarding oral mucosa regeneration, Wang and colleagues [12] showed tissue engineering strategies using the electrospinning process. Such devices were applied in porcine, rabbits and dog models of oral mucositis induced by anticancer therapies. As there are few studies reporting the use of electrospun fibers for cell therapy to treat oral diseases [12], the present work intends to show how an electrospun PLGA construct, with the addition of fibroblast-like cells from hamster cheek paunch, can serve as a future dressing for chemotherapy-induced oral mucositis ulcerations in hamster model. 

It is noteworthy the great potential of this technology to produce nanofibers or microfibers using an electric field. This field can be tunable to control the deposition of aligned or random micro or nanofibers onto a collector, in a 3D pattern, forming a biomimetic scaffold for tissue engineering applications [12]. In general, this system regards advantages, such as the possibility of obtaining a large surface area in relation to the volume, organized interfibrillar spaces, controlled malleability and the facility in generating materials with a variety of sizes, texture and shapes [13]. In addition, 3D electrospun materials keep properties that promote the flow of nutrients from the biological microenvironment, such as oxygenation and vessel penetration throughout the device, accurately as it happens in the human organism [12]. As a result, it runs accelerated cells proliferation [14]. 

Recently it was shown that, when covered with nanoparticles or bioactive factors, PLGA membranes can turn into a promising bifunctional scaffold [4]. In this way, those scaffolds play a crucial role in cell migration and differentiation, both critical for their future applications in the field of tissue engineering and regenerative medicine. Moreover, medical devices must be sterilized before animals or human applications. Although a variety of techniques can be used to sterilize medical devices, our group previously demonstrated, by physicochemical and structural analysis, that 3D electrospun PLGA membranes treated with gamma radiation showed preserved structures [15].

Over 50 years the term “biocompatibility” has been widely used to qualify the biological properties of a biomaterial. For instance, Liang et al. [16] reported that the selection and fabrications of biomaterials with different structures and forms, such as films, hydrogels, electrospun scaffolds, sponges and foams should be tested to enhance therapeutic efficiency. Accordingly, the present work poses special emphasis on in vitro tests to study cell–biomaterial interactions using MDCK cell-line and primary fibroblast-like cells isolated from hamster cheek paunch with manufactured electrospun PLGA membranes. Moreover, in the present study, tissue-biomaterial immune responses were analyzed after in vivo hamster implantation over time. To our knowledge, this is the first time that 3D electrospun PLGA membranes are studied in a hamster model.

Madin-Darby canine kidney (MDCK) cell-lines are epithelial cells used in experimental research models due to their potential to adhere and form either tubules or flat monolayer sheets in 3D and 2D structures, respectively. This multicellular architecture is formed in virtue of either intercellular interaction between epithelial cells or upon cell–matrix interactions in the appropriate scaffold [17]. On the other hand, primary fibroblast-like cells from hamster’ cheek pouch were chosen by our group due to our future tissue engineering challenges, to test this construct in hamster models [18]. 

Biological responses of cell–material interactions are a function not only of biomaterials design or their intrinsic properties but also of the microenvironment. Specific and long-term reactions can differ as a function of the organ, tissue and or species [4,19,20,21,22]. In this milieu, the material’s surface properties play an important role in modulating the immune system response, such as the foreign body reactions [23,24]. This immunological response has been analyzed in experimental models as being a natural response of the organism, which will remain as long as polymer fragments are present at tissue–material interfaces [25,26]. Such response recognized as foreign body response (FBR) is classified into two main types of multinucleated giant cells, the Langhans-type giant cells (LCs) and the multinucleated-type giant cells (MNGCs). The LC is present within granulomas of infectious and non-infectious etiology. Morphologically, LCs exhibit a circular or ovoid shape with nuclei arranged in the periphery of the cells, in a limited number (10–20 nuclei per cell), adopting a circular or horse-shoe pattern [27]. On the contrary, MNGCs result from the macrophage response to indigestible substances, exhibiting an irregularly shaped cytoplasm, which may contain hundreds of nuclei per cell [27].

Although PLGA is a widely used polymer in the area of regenerative medicine and tissue engineering, its processing in porous membrane thanks to electrospinning can modify the inflammatory reaction that it will induce and therefore its interaction with tissues [12]. The present study focused on the in vitro morphological analysis of cell–material interactions and on the in vivo immune response after PLGA electrospun membrane implantation. To process the aforementioned analysis over time, our group applied morphological, cytochemical and immunohistochemical approaches associated to light (conventional optical and laser scanning confocal) and electron (scanning and transmission) microscopy. These tools revealed cell proliferation over time onto the PLGA membranes in vitro, and tissue immune responses after membrane implantation, without fibrous tissue formation at the biomaterial interface.

## 2. Materials and Methods

### 2.1. PLGA

PLGA 85% L-lactide and 15% glycolide (*w*/*v*), (Purac: PLG 8531-lot No.1504000089, Amsterdam, The Netherlands) were produced using the randomly oriented electrospinning process. Polymeric solutions were prepared dissolving 5% PLGA (weight/volume) in the solvent-binary of chloroform (CHCl3-Vetec, Rio de Janeiro, Brazil) and dimethylformamide (DMF-Vetec, Rio de Janeiro, Brazil) systems (80/20 *v*/*v*), and magnetically stirred at room temperature.

### 2.2. Electrospinning Process

The polymeric solution was introduced into a syringe with a needle attached to an ejector pump under the voltage of 15 kV using the Glassman PS/FC 60p02.0-11 (KDS 100 Infusion Syringe Pump) high voltage source. The solution was ejected induced by the potential difference between the needle tip and the screen, with a working distance of 15 cm and a feeding rating of 0.6 mL/h for 6 h. PLGA filaments were deposited onto an aluminum foil and collected as a membrane. The process was standardized for all frameworks produced (dimensions: 10 × 10 cm^2^ and thickness of 6.6 µm, analyzed by scanning electron microscopy). Next, PLGA membranes were sterilized using gamma radiation (Cobalt 60 source—MDS Nordion, model GC 220 E, Ottawa, ON, Canada), with a total dose of 15 KGy [28].

### 2.3. In Vitro Experiments

#### 2.3.1. MDCK Cells Cultured onto PLGA Membranes

Madin-Darby canine kidney cells (MDCK, ATCC, cat. No. PTA-6503) were plated in 25 mm^2^ culture bottles, in complete Cascade Medium 200 (M200-500) culture medium, incubated at 37 °C, in an atmosphere of 5% CO_2_ and 95% humidity. After confluence, cells were trypsinized (2 mL TrypLE ™ Express Enzyme (1×), phenol red, cat. No.12605036-Gibco^®^) for 5 min. Complete culture medium (2 mL) was used to inactivate the process. Next, cells were centrifuged at 700 g for 3 min. PLGA samples (1 × 1 cm^2^) were glued on the bottom of the wells of a 6 wells cell culture dish, covered or not with 20µg/mL fibronectin (human plasma fibronectin, Sigma-Aldrich, St. Louis, MI, USA, cat. No. F0895) in culture medium. After 1 h, PLGA samples were washed twice with sterile phosphate saline buffer (PBS) pH 7.4. Next, 1 mL culture medium containing MDCK cells (10^4^) was applied only onto the surface of each membrane, and incubated at 37 °C, in an atmosphere of 5% CO_2_ and 95% humidity for 1 h. After 1 h, complete culture medium (2 mL) was added in the well to cover each sample, and changed every 48 h. The experiments were carried out in quadruplicate and analyzed after 3, 6 and 12 days. PLGA membranes were fixed using 4% buffered paraformaldehyde solution for 15 min, washed in PBS, pH 7.4, twice; permeabilized with triton X-100 (0,1%) in PBS for 15 min; washed again in PBS, and incubated in PBS containing 0.5% serum albumin (BSA) for 1 h. Samples were treated with Alexa Fluor^®^ 555 phalloidin 1:100 (Invitrogen, cat. No. A34055-Thermo-Fisher Scientific, Waltham, MA, USA), in PBS-0.3% BSA, and stored protected from light for 1.5 h. Next, samples were washed with PBS and treated with Hoechst solution (Bisbenzimide H33342 trihydrochloride solution in PBS, 1:1000, Sigma-Aldrich, cat. No. 14533) for 30 min, washed with PBS (3 times). Confocal microscopy analysis (LSM 700, Zeiss) was performed using the objective lens (W Plan-Apochromatic 63×/1.0 M27). Ten fields per sample were obtained over time, using the X-Z orthogonal axis and captured in the Image J Fiji software [29], to count the cell nuclei. Mean ± sd was used for analysis over time. The difference in cellularity was verified using the GraphPad Prism 5 program. For scanning electron microscopy (SEM) analysis, each sample was fixed in a solution of 1 mL glutaraldehyde 2.5% (Reagen—Quimibras Industrias Químicas S.A., Rio de Janeiro, Brasil); 1 mL of 4% formaldehyde (Merck) and 2 mL of Sorensen phosphate buffer (disodium hydrogen phosphate-Na2HPO4—11.876/L; potassium di-hydrogen phosphate (KH_2_PO_4_) at 0.2 M, pH 7.2, for 1 h. Then, the samples were washed in the same buffer 3× for 10 min, dehydrated in serial ethanol (Merck) solutions (30, 50, 70 and 90%) for 10 min each, ending with absolute ethanol, twice for 20 min and dried in ambient air. Then, the samples were fixed onto a metallic support with carbon tape, metallized by gold sputtering (20 nm) and observed in a FEI Quanta 400 scanning electron microscope.

#### 2.3.2. Primary Hamsters’ Fibroblasts Cultured onto PLGA Membranes

Primary fibroblasts from left hamsters’ cheek-pouch were isolated after surgery using intraperitoneal ketamine (80 mg/kg) and xylazine (60 mg/kg). The excised cheek pouch was washed with povidine-iodine for 5 min, next rinsed with a saline solution 1 min, treated with nystatin (100.000 UI/Laboratorio Teuto S/A-Brasil) for 5 min, and finally immersed in sterile saline solution for 5 min. Tissues were sectioned into small pieces under a laminar flow hood (Vertical laminar flow—Pachane LTDA, model PA 320, No.03201. Pirasicaba, Brasil); immersed in collagenase type 1 solution (1 mg/mL-Gibco/Life Technologies) in culture medium (100 mL) inside an incubator at 37 °C under agitation for 3 h and inactivated using fetal bovine serum (10%—Vitrocell/lot 014/18). Finally, the cells were isolated by filtration (cell strainer 100 μm/Sigma-Aldrich) and centrifugation (700 g) of the supernatant. The pellet was suspended in 1 mL low glucose DMEM culture medium, supplemented with 10% fetal bovine serum, 100 IU/mL penicillin, 100 mg/mL streptomycin and 0.01 mg/mL amphotericin (Sigma Aldrich, St. Louis, MO, USA), plated in a 25 cm^2^ culture bottle containing 4 mL culture medium, and kept in an incubator at 37 °C in a 5% CO_2_ atmosphere. Culture medium was changed every 48 h after 3 washes with PBS, and stored at 37 °C in a 5% CO_2_ atmosphere for 4 days. In another assay PLGA membranes (1 × 1 cm^2^) were fixed with silicone glue and treated with 100 µL of a mouse collagen solution (Sigma-Aldrich) 10% (*v*/*v*) in serum-free DMEM, incubated for 1 h in the laminar flow and washed using PBS. Next, a suspension of 10^4^ or 3 × 10^5^ cells in 100µL of complete DMEM culture medium was added to each membrane and cultured over 1, 3, 6 and 12 days. Over time, the membranes were fixed in 4% paraformaldehyde in PBS for 30 min, washed in PBS for 5 min 3× and kept in a 20 mL Falcon tube with PBS at 4 °C. After 12 days, all the membranes with cells were permeabilized with Triton X-100 (0.1%) in PBS for 10 min, washed 3× in PBS for 5 min, immersed in PBS containing 0.5% BSA for 1 h, washed in PBS for 5 min and incubated with phalloidin-Alexa Fluor 488 (Invitrogen, 1:80) in PBS containing 1% BSA, for 1 h; washed 3× in PBS and finally stained with Hoechst solution (1: 1000 in PBS/Sigma-Aldrich) for 30 min. After washing 3× in PBS for 5 min, membranes were mounted on 3 cm diameter imaging dish with a coverslip. The side of cell seeding was fixed facing the bottom of the coverslips, and covered with a new drop of mounting medium to fix another coverslip. The cell nuclei were counted on the confocal microscope (10 fields per membrane) (LEICA-TCS-SPE, inverted O DMi8, using the ACS-APO 63×/1.30 lens). For SEM analysis, membranes were fixed in a solution of 2.5% glutaraldehyde, 4% paraformaldehyde and 0.1 M sodium cacodylate buffer, for 2 h at room temperature, washed with the same buffer for 10 min, post-fixed in 1% buffered osmium tetroxide solution for 40 min, washed with distilled water for 10 min, dehydrated in serial ethanol solutions (30%, 50%, 70% and 90%) for 10 min each and 100% ethanol for 20 min twice, at the end. Membranes were immersed in hexadimethylsilazane in absolute ethanol (*v*/*v*) solution for 5 min twice and dried. Next, the membranes were mounted on carbon strips attached to a metal plate and metallized with gold (20 nm) (BAL-TEC SCD 050) for analysis (SEM FEI QUANTA 250, operated at 15 kV).

### 2.4. In Vivo Experiment

Golden Syrian hamsters were purchased at Oswaldo Cruz Foundation-Rio de Janeiro, Brazil, at six weeks of age and maintained at the Health Science Center bioterium of Federal University of Rio de Janeiro-UFRJ-Brazil, in environmental, nutritional, and health-controlled conditions, according to “The Guide for Care and Use of Laboratory Animals” (DHHS Publication No (NIH) 85-23, Office of Science and Health Reports, Bethesda, MD 20892—available at: http://www.nih.gov). The Ethics Committee on the Use of Animals in Scientific Experimentation at the Health Sciences Center of the Federal University of Rio de Janeiro, registered in the National Council for Animal Experimentation Control (CONCEA-Brazil), certified the use of hamsters in the present study (protocol No. 003/15, approved on 15 April 2015). Animals between 120 and 205 g were included in the present study. The animals were kept in appropriate cages containing 3 animals/cage, lined with sterile shavings, under constant temperature (23 ± 2 °C), in the standard light/dark cycle (12/12 h), with unrestricted access to water and feed. After the experiments animals were euthanized according to the “American Veterinary Medical Association Guidelines on Euthanasia”, 2007 (available at http://www.nih.gov).

### 2.5. PLGA Implantation and Material Collection

PLGA membranes (1 × 1 cm^2^) were implanted in the dorsal region of the hamsters over time (7, 15, 30 and 90 days). The animals were randomly separated in 4 groups and were anesthetized with Ketamine (80 mg/Kg) and Xilasina (60 mg/Kg), in the intraperitoneal region (i.p.) for surgical procedures of PLGA implantation. PLGA was implanted in the upper dorsal region, after asepsis using 70% alcohol and shaving. Two PLGA samples (1 × 1 cm^2^) were implanted in the subcutaneous upper dorsal region, one in each side. Next, the skin edges were sutured (5.0 nylon-Technofio). The animals were kept under observation for 24 h to control post-surgical behavior. Over time (7, 15, 30 and 90 days), PLGA samples were surgically removed with the adjacent skin, and divided in two pieces for morphological analysis. Next, the animals were euthanized, by inhaling CO_2_ followed by decapitation through a guillotine.

### 2.6. Sample Preparation for Morphological Analyzes by Light and Electron Microscopies

The implanted materials were fixed in 4% paraformaldehyde in PBS, pH 7.4 at 4 °C for 2 h. Next, washed in running water, dehydrated in serial solutions of ethanol from 30% to 100% twice for 20 min, clarified in xylene (2 baths of 30 min) and embedded in paraffin. Sections of 5 μm-thick were performed on a rotary microtome (Leica Microsystem RM 2125^®^, Wetzlar, Germany) and collected on glass slides. Dried histological sections were stained with hematoxylin-eosin (HE) and Masson’s trichrome stains. The other samples were immersed in Karnovsky’s fixative solution (4% paraformaldehyde solution, 2.5% glutaraldehyde and 0.1 M sodium cacodylate) for 2 h, washed 3 times in the same buffer, cleaved in cubes of about 1 mm^2^ and post-fixed with 0.1 M osmium tetroxide solution for 30 min. Next, they were washed with cacodylate buffer 3 times for 10 min, dehydrated in serial ethanol solutions to absolute 2 times for 20 min and embedded in Spurr resin. The selected region was cut into 90 nm thickness and contrasted with 1% uranyl acetate and 2% lead citrate for transmission electron microscopy analysis in a Morgagni 268 (FEI company, Hillsborough, OR, USA) operated at 80 KV.

### 2.7. Immunohistochemistry

Paraffin sections were collected on silanized histological slides (Sakura Finetek, Staufen, Germany) for immunohistochemistry. After adhesion, histological sections were dewaxed in xylene and hydrated. The samples were washed with 50 mM ammonium chloride solution, in phosphate buffered saline (PBS) pH 8.0, for 15 min, to block free aldehyde residues and washed in PBS; permeabilized with Triton X100 (0.5%) in PBS for 15 min, followed by a bath containing 0.3% hydrogen-peroxide in methanol to inhibit endogenous peroxidase for 15 min, in the dark. After washing with PBS, pH 7.4, sections were submitted to heat-mediated antigen retrieval in either the microwave (potency 800 W) or steamer, according to the antibody used (Table 1). After cooling, the histological sections were incubated with PBS containing 5% bovine serum albumin (BSA), normal 5% goat serum (1 h) in a humid chamber, at room temperature, and then, primary antibodies (Table 1) were incubated. Sections were maintained in a humid chamber, for about 20 h, in the refrigerator. Afterwards, the sections were washed in a PBS solution containing 0.25% Tween-20 (PBS-Tween), followed by incubation with the secondary antibody conjugated to peroxidase (Envision ^TM^ Dual link system-HRP—cat. No. K4601, Dako, CA, USA), for 1 h, followed by washes in PBS-Tween. Peroxidase was developed with the chromogenic substrate diaminobenzidine (Liquid DAB, Dako, cat. No. K3468), followed by washes in PBS-Tween and distilled water and assembly of sections between slide and coverglass with Entellan^®^. Negative controls were performed by incubating the histological sections with non-immune rabbit or mouse serum or with the antibody diluent in place of the primary antibody.

### 2.8. Morphometrical and Statistical Analysis

Semi-quantitative analyses of histological sections were performed using an Eclipse E800 light microscope (Nikon, Japan) coupled to a digital camera (Evolution VR Cooled Color 13 bits (Media Cybernetics, Bethesda, Rockville, MD, USA) using the 2×, 4×, 10×, 20× and 40× objectives lens. The interface capture used was Q-Capture 2.95.0, version 2.0.5 (Silicon GraphicsInc, Milpitas, CA, USA), and the images were transmitted to a color LCD monitor, captured (2048 × 1536 pixel buffer) in TIFF format and digitized. The images were captured after calibration of the appropriate color and contrast parameters and remained constant for each type of staining or immunohistochemistry. The quantification of CD3^+^ cells (T lymphocytes) or myeloperoxidase + (neutrophils), were performed on the captured images (10 microscopic fields of the slides stained with the respective antibodies, using the 40× objective lens). The results were expressed as number of reactive cells/histological field ± standard error of the mean. The amount of multinucleated giant cells and capillaries (transversal sections) were calculated on 20 photomicrographs randomly obtained from HE stained histological sections (objective lens 20×). Results were expressed as number of multinucleated giant cells or number of capillaries, /histological field ± standard error of the mean.

For statistics, the in vitro analysis of cells was performed by one investigator (AC). Cells nuclei were counted on Image J Fiji software or in Image ProPlus 5.0 (Media Cybernetics, MD, USA). For the in vivo quantification, one investigator (pathologist, CMT) acquired all histological images and performed the quantification, in a blinded manner. The data were analyzed using GraphPad Prism 5.0 software (GraphPad Software, Inc., La Jolla, CA, USA). The differences between the groups were analyzed using one-way ANOVA on ranks (Kruskal–Wallis test), followed by a post hoc test (Dunn’s multiple comparison test or Neuman–Keuls). Differences were considered significant at *p* < 0.05, with asterisks indicating the level of significance: * *p* < 0.05, ** *p* < 0.01, *** *p* < 0.001; ns indicates no significant differences.

## 3. Results

### 3.1. In Vitro Experiments 

#### 3.1.1. MDCK-Cell Line

MDCK cell line is a widely used epithelial cell line to test the monolayer formation. They were used in the present work to test the electrospun PLGA (85:15) membranes produced herein. The cells adhered and proliferated on both fibronectin-coated (FN) and uncoated (no-FN) PLGA membranes, independently, over time. These results showed that electrospun membranes with random fibers, with or without FN promoted cell proliferation, such as shown in Figure 1A–C, with cells forming a monolayer onto the membranes. MDCK cells preferentially adhered and proliferated on the fibronectin (FN) coated PLGA membranes as seen in (Figure 1C). After 3 days in culture, the number of epithelial MDCK cells on FN-coated or no-FN membranes showed no difference (Figure 1C: 3 days FN: 27 ± 2 cells/ mm^2^ to specify; 3 days no-FN: 24 ± 1 cells/ mm^2^, *n* = 4). An increase in the number of cells was observed after 6 and 12 days (Figure 1C), regardless of treatment with fibronectin. However, after 12 days, there was a greater number of cells on FN-coated membranes in comparison to no-FN membranes (Figure 1C: 6 days FN: 59 ± 8 cells/mm^2^; no-FN: 50 ± 8 cells/mm^2^; 12 days FN: 131 ± 16 cells/mm^2^; no-FN: 102 ± 16 cells/cm^2^).

To illustrate, a confocal image is showing MDCK cells through the fibers of no-FN membrane fibers (Figure 1(Aa)) after 3 days in culture. For comparison, image of FN-coated membrane (Figure 1(Af)) is showing cells proliferating through the fibers after 3 days in culture (thick arrow). In an equivalent view of Figure 1(Aa,Af), the cell nuclei (Figure 1(Ab,Ag)) are exhibiting different distribution, showing that cells are following the random design of the fibers. In another equivalent view of A-a (Figure 1(Ac)), confocal image of no-FN membrane (Figure 1(Ac), arrows) after phalloidin staining after 3 days reveals that cells are distributed among the fibers but are not connected since no circumferential actin belt is visible at that time (Figure 1(Ac), circle). For comparison, confocal image of FN-coated membrane (Figure 1(Ah)) exhibits regions where cells appeared to be connected since circumferential actin belt is sometimes visible (Figure 1(Ah), thin arrows) among cells without adhesions (Figure 1(Ah), thick arrows), still not taking the classic cuboidal morphology after 3 days. Image of orthogonal projections (Figure 1(Ad,Ai)) showing cells after 3 days reveals the monolayer formation onto no-FN (Figure 1(Ad)) or FN-coated membranes (Figure 1(Ai)), both with few in-depth cell migration after 3 days. 

Image of cells onto no-FN membrane after 12 days (Figure 1(Ba,Bf)) reveals more localized MDCK cells among PLGA fibers (Figure 1(Ba), arrow; double arrow) in comparison to FN-coated membrane showing a dense monolayer of cells among the fibers (Figure 1(Bf), double arrow). An equivalent image of Figure 1(Ba), f reveals the cells’ nuclei on no-FN and FN coated membranes (Figure 1(Bb,Bg)), respectively in close contact to each other and exhibiting different directions according to fibers’ organization. Moreover, images after phalloidin staining of no-FN or FN-coated membranes (Figure 1(Bc,Bh)), depicts increased cell connections with circumferential actin belt; both taking their classic cuboidal morphology after 12 days (Figure 1(Bc,Bh), arrows). Note that orthogonal projections images after 12 days (Figure 1(Bd,Bi)) show MDCK cells taking decreased size in comparison to cells after 3 days in culture. This can be explained by the proliferation of the cells, causing them to thicken and take the classic cuboidal morphology of epithelial cells at confluence, visible on orthogonal projections. SEM images of MDCK cells reveals the presence of isolated cells after 3 days (Figure 1(Ae,Aj), thin arrows and asterisks), exhibiting less amount of MDCK cells onto no-FN and FN membranes, in comparison with the cluster of cells with extracellular matrix visible after 12 days (Figure 1(Be,Bj), white arrows and asterisks). In addition, SEM image of FN-coated membrane after 12 days (Figure 1(Bj), white arrows) shows cells spreading in close contact with the membrane fibers’, by forming cell extensions towards areas without cells following PLGA fibers direction. 

Number of cells/mm^2^ over time is illustrated in Figure 1C. The results are expressed as mean ± SEM (standard error of the mean). Images of cell nuclei (*n* = 10) stained with Hoechst on each sample (*n* = 4) were captured by CLSM (63× objective lens) and counted. Data were submitted to one-way ANOVA followed by Bonferroni’s multiple comparison test (ns: not significant, ** *p* < 0.001, *** *p* < 0.0001). 

#### 3.1.2. In Vitro Experiment Using Primary Fibroblast-like Cells

Primary fibroblasts isolated from hamsters’ cheek paunch were seeded onto collagen-coated electrospun PLGA (85:15) membranes. To select the best construct, two cell concentrations (10^4^ or 3 × 10^5^ cells) were tested and compared over time (1, 3, 6 and 12 days) using confocal microscopy (CLSM). Both concentrations of cells adhered and proliferated over time onto PLGA-coated membranes. The construct with 3 × 10^5^ cells showed increased cell proliferation after 6 and 12 days in culture; such cells took an elongated morphology and migrated towards the interior of PLGA membrane, posing an ideal dressing for tissue engineering and future tests as cell delivery therapy.

The primary culture of fibroblast-like cells is represented on a phase-contrast image (Figure 2A) (fourth passage) after 4 days, and the random electrospun PLGA fibers covered with collagen is represented on a CLSM image (Figure 2B). After 1 day in culture, cells adhered on the surface of the fibers (Figure 2C, white arrows depict the PLGA fibers), and took an elongated morphology, taking different directions following the fibers (Figure 2D, white arrows). The maximum projection confocal images show cell proliferation (3 × 10^5^) over time and their respective in depth migration (Figure 2(Ee,Ff,Gg,Hh)). After one day, cells were only on the surface of the membrane (Figure 2(Ee)). After 3 days (Figure 2(Ff), white arrows) confocal image shows cells initiating the in-depth migration. After 6 days (Figure 2(Gg), white arrows) more cells were migrating to the interior of the membrane, while a deeper in-depth migration was observed after 12 days (Figure 2(Hh), white arrows), in three different areas of the same membrane. Results of cell proliferation in the assay with 3 × 10^5^ cells after 6 and 12 days, reveal significantly higher proliferation of cells compared to other times (Figure 2I). However, it was possible to observe an increase in the number of cells in each assay with 10^4^ cells (1 day: 9 ± 2; 3 days: 30 ± 2; 6 days: 43 ± 2; 12 days: 47 ± 4; or 3 × 10^5^ cells) or 3 × 10^5^ cells (1 day: 35 ± 4; 3 days: 45 ± 10; 6 days: 64 ± 3; 12 days: 76 ± 2) (Figure 2J). 

SEM images of collagen-coated membranes showed a collagen network in the interfibrillar spaces (Figure 2K, fiber: arrow; collagen: asterisk). In another SEM image (Figure 2L), a cluster of cells in extracellular matrix spread on the surface of the PLGA membrane in intimate contact with PLGA fibers (black arrows). Interestingly, sometimes cells showed an elongated morphology following PLGA fibers (Figure 2M, arrow heads), while other cells display more spread morphologies in the interfibrillar space when adhering to multiple fibers (Figure 2M, asterisks).

### 3.2. In Vivo Experiment

#### 3.2.1. Histopathology

The skin of hamsters is constituted by the epidermis, dermis (papillary and reticular layers), hypodermis and a skeletal muscle layer beneath the hypodermis. After PLGA implantation, the inflammatory response induced by PLGA membranes was a well localized chronic inflammatory response in the implantation bed of the membranes, below the muscular layer, remaining during all period of observation. The histological features of the foreign body response induced by the PLGA membrane are represented in Figure 3A–L. The inflammatory response delimited the biomaterial, forming an interface with the adjacent connective tissue without presenting a fibrotic capsule in any timeline (Figure 3A–L), and was constituted by a foreign body reaction consisting mainly by mononuclear cells of the monocytic lineage including macrophages, epithelioid cells and multinucleate giant cells (Figure 3A–L). The membranes were located inside the inflammatory site, such as verified with polarizing microscopy, represented by semi-thin sections after 7 days (Figure 3A,B), and after 15 days (Figure 3D,E) post implantation (PI). Mononuclear cells (Figure 3C) and multinucleate giant cells (MNGCs) (Figure 3C yellow arrows) surrounded the membrane after 7 days PI; and a foreign body response (FBR) beneath the muscle layer was observed after 15 days PI (Figure 3F, arrow). Another histological image after 15 days PI is showing the FBR constituted Langhans-type multinucleate cells (LCs) (Figure 3G arrows), lymphocytes (Figure 3G square) and epithelioid cells (arrow) with an elongated shape in rows surrounding the membrane (Figure 3H arrows). After 30 days histological image shows the FBR formed as a well delimited lesion in the hypodermis (Figure 3I arrow), with bands of connective tissue (Figure 3J, thick arrows) and capillaries (Figure 3J, thin arrow) present inside the lesion. After 90 days PI, histological image shows persistence of the inflammation (Figure 3K, arrow) and presence of both MNGCs (Figure 3L, thin arrow) and LCs (Figure 3L, thick arrow).

The LCs (multiple nuclei disposed at the periphery of the cell) (Figure 4A, black arrows) increased in number from 7 to 30 days PI decreasing successively up to 90 days PI (Figure 4B). Associated to LCs, capillaries sections were present in high numbers at day 7 PI, decreasing along time (Figure 4C,D). A histological image of a region showing the presence of capillaries (Figure 4C, black arrows) along the evolution of the implanted PLGA (Figure 4C, asterisk) represents the PLGA-induced inflammatory reaction. The quantification of the number of capillaries is represented by the number of capillaries over time, showing that the number of capillaries decreased up to 90 PI (Figure 4D). Number of capillaries (transversal sections) were calculated on 20 photomicrographs randomly obtained from HE stained histological sections (objective lens 20×). Results were expressed as number of capillaries, /histological field ± standard error of the mean. 

#### 3.2.2. Immunohistochemistry and Morphometry

The inflammatory process was analyzed over time after PLGA membranes implantation. For this goal, the CD3 antigen was used to measure the total number of T cells, and the myeloperoxidase (MPO) reactivity for polymorphonuclear neutrophils (PMNs). Both immunohistochemistry assays quantified these cellular populations, which decreased up to 90 days, showing that the inflammatory process decreased along PLGA fragments degradation.

In the analysis, the total number of T cells (Figure 5A,C,E,G) showed variations over time. The number of T cells increased after 7 and 30 days PI and decreased after 15 and 90 days PI (*p* = 0.0002) (Figure 5A,C,E,G,I), while neutrophils (myeloperoxidase+ cells) levels were significantly elevated at the implantation bed after 15 days PI (*p* < 0.05), decreasing after 90 days PI (*p* = 0.004) (Figure 5B,D,F,H,J). Results are expressed as the number of CD3 or myeloperoxidase^+^ cells/histological field ± SEM (standard error of the mean). 

#### 3.2.3. Transmission Electron Microscopy (TEM)

In this work we chose a high-resolution imaging technique of TEM to show detailed images of PLGA fragments in intimate contact with the cells over time. TEM images showed various populations of macrophages, epithelioid cells and multinucleated giant cells at the granulomatous inflammatory process induced by the PLGA membranes between 7 and 90 days PI (Figure 6A–H). The inflammatory process was observed in the periphery of PLGA, showing mononuclear cells forming a foreign body reaction (FBR) in contact with the extracellular matrix (ECM) 7 days PI (Figure 6A). In the periphery of PLGA, TEM image is showing PLGA fragments (Figure 6B, arrows) in contact with a multinucleate giant cell (Figure 6B(GC)) 7 days PI. Another TEM image shows PLGA fragments in contact with epithelioid cells (EP) (Figure 6C(EP)), presenting different morphologies, with prominent cytoplasmic extensions such as filopodia following the topography of PLGA fragments (Figure 6C, arrows) 7 days PI. PLGA fragments were found inside phagosomes (Figure 6D arrow), in the interior of the cell (Figure 6D, asterisk) or in the extracellular space (Figure 6E, arrow) 15 days PI. However, an interesting and rare light microscopy image shows LCs containing material inside (Figure 6F, arrows) 30 days PI, such as depicted by TEM images showing PLGA fragments in the interior of cells (Figure 6G, arrows) 30 days PI. Finally, TEM image 90 days PI show decreased fragments of PLGA between the cells (Figure 6H, arrows).

## 4. Discussion

Poly (lactic-co-glycolic acid) (PLGA) is a synthetically manufactured linear copolymer constituted by different proportions of monomeric lactic acid (LA) and glycolic acid (GA). It is used in bioengineering as scaffolds for corneal and skin tissue engineering [30], as well as for bone regeneration [31,32,33] or for drug delivery [33,34]. PLGA presents a good biocompatibility, excellent mechanical properties, controllable biodegradability and electrospinnability. Herein, we are describing cell-interactions with electrospun PLGA (85:15) membranes in vitro (MDCK cell line or primary culture of hamster oral fibroblasts) or in vivo when implanted in the immunocompetent hamster subcutaneous tissue.

Recently, our group analyzed the physico-chemical properties of the electrospun PLGA membranes used in the present study [15]. The membranes were mostly constituted of fibers less than 1 μm in diameter [15]. In fact, biodegradable electrospun membranes constituted by thin fibers have been shown to promote cell adhesion, migration and proliferation [11,29,35]. In this sense, our team arouses the hypotheses that the fibronectin-coated electrospun PLGA membranes used in the present study should contribute to cell proliferation due to its biomimetic properties, since it has been shown in a previous study that gingival fibroblasts can adhere and proliferate when seeded onto collagen or PLGA substrates [36].

To achieve better results, PLGA hybrid membranes containing collagen or fibronectin have been manufactured [37,38,39,40]. Indeed, human dermal fibroblasts cultured onto collagen-coated electrospun PLGA substrates secreted extracellular matrix (ECM) on the substrate forming a fibrillar network in the PLGA interfiber spaces, promoting better cell–material interactions [38]. Our data showed similarities with the study of Sadeghi et al., [38] regarding collagen I-coated PLGA membranes, which promoted fibroblast-like cell proliferation over time and secreted ECM onto the surface of the scaffold. Inanç et al. [41] also showed that periodontal ligament cells seeded onto PLGA membranes exhibiting the same proportions of monomers as herein, produced collagen I and fibronectin which covered the PLGA fibers [40]. Considering the average length of a fibroblast, Lowery et al., [42] demonstrated that membranes possessing interfibrillar spaces above 6 μm and less than 20 μm are permissive to the fibroblast adhesion, locomotion and proliferation inside the scaffold, while this is not the case when the interfibrillar spaces are higher than 20 μm. Although in the present study the interfibrillar spaces were not measured, it can be hypothesized that the net of collagen I formed in the interfibrillar spaces after membrane bioactivation, can favor fibroblast-like cells proliferation over time.

The MDCK cells are a model of canine kidney cell line used in drug screening and biomedical research [43,44]. Moreover, these cells have been widely used in a variety of research studies using synthetic polymers, regardless of polymers fabrication or formats, due to their ability to follow different topographies. In this sense, we appreciated the migration behavior of MDCK cells onto 3D electrospun PLGA scaffolds over time. Indeed, since the 3D electrospun PLGA scaffold used herein possesses randomly oriented fiber topography, conferring interconnected fiber-forming porous, it was considered an ideal architecture to appreciate the monolayer and epithelial barrier formation of these cells. Our in vitro data confirmed that MDCK cells also proliferated onto the PLGA membranes coated or not with fibronectin after three and twelve days, independently. Comparing the two conditions, cell proliferation showed significant better results on the fibronectin-coated PLGA membranes after 12 days. Accordingly, we showed a cluster of cells taking on a classical cuboidal morphology and forming a confluent monolayer with cell polarity after 12 days. 

Our morphological study revealed that the implanted membrane promoted a highly cellular foreign body response up to 90 days PI, forming at all studied times a well-delimited structure in the site of implantation. The implantation of biomaterials in vivo frequently elicits a chronic inflammation at the tissue-implant interface followed by wound healing responses and tissue fibrosis [45,46,47]. Concomitant with the formation of macrophage-derived foreign body multinucleated giant cells, constituting LCs or MNGCs, granulation tissue develops around the biomaterial. Factors secreted by macrophages adhering onto the surface of the biomaterial and the multinucleated giant cells formations attract fibroblasts and transform them into myofibroblasts [48,49]. These cells become aligned around the biomaterial and deposit great amount of collagen and other proteins creating a fibrous capsule around the implanted biomaterial. The fibrous encapsulation is considered as the end stage of the foreign body reaction and healing response to a biomaterial.

In the majority of cases, biomaterials promote the progression of such sterile inflammation to a foreign body response [47]. In the late 1980s and early 1990s, the occurrence of a typical nonspecific foreign-body reaction around internal fracture fixation implants made of pure polyglycolide or polylactide and even of glycolide-lactide copolymer was reported [50,51,52,53,54].

The foreign body response induced by our manufactured electrospun PLGA membranes was extremely cellular with a high amount of LCs derived from fusogenic macrophages. Macrophages are extremely versatile and plastic, being capable of adhering and recognizing foreign materials. At this time, they show typically a classically activated phenotype secreting inflammatory cytokines, ROS, protons release and degradative enzymes. Besides that, they display high phagocytic capacity, being able to phagocytose particles up to a size of 5 mm. In the presence of particles larger than 5 mm, macrophages coalesce to form foreign body multinucleated giant cells [55,56]. At this time, these cells display a reduced phagocytic activity, but an enhanced degradative capacity [56] at the expense of protons secretion, enzymes and ROS [57,58]. Kim [59] reported three steps on the biological pathways of local tissue responses to biomaterials: (i) organization of the inflammatory responses; (ii) monocytes migration to implantation site which differentiate into macrophages and fibrous capsule development and (iii) rapid degradation of the polymer and enhanced formation of fibrous tissues generated in the second step. These biological steps can guide the comprehension and analysis of the immunological responses in different hosts. In this sense, macrophages orchestrate the FBR that remain at the biomaterial/tissue interface when they do not succeed in phagocytizing the material [43,60]. We showed using histological and immunohistochemistry study that the electrospun PLGA membranes induced the recruitment of cells of the monocyte/macrophage lineage with the formation of epithelioid cells, LCs, as well as transitional cells which were similar to the cells seen after the implantation of fragments of MELINEX TM plastic in the subcutaneous tissue of rats [61,62]. Our ultrastructural study using TEM showed the difficulty of identifying these cell types since the transformation of monocytes to macrophages, macrophages into epithelioid cells or MNGCs were gradual, without a clear-cut distinction between the different cell types. Foreign body giant cells were recognized by the numerous nuclei inside cytoplasm, while macrophages, epithelioid cells and even giant cells, showed in the cell surface numerous invaginations and finger-like projections and numerous micropinocytotic vesicles. In addition, some cells mainly in the later stages (after 30 days PI) presented numerous filopodia, which surrounded the PLGA, or phagosomes, some of them containing fragments of the PLGA. The presence of phagosomes containing material similar to that of PLGA fibers could be correlated with the vacuoles seen in light microscopy inside some cells, suggesting that these cells have some capacity for phagocytosis. According to our knowledge, light microscopy images showing cells internalizing PLGA fragments were not found in the literature till now.

Neutrophils are present within the first days (up to day 2) after the biomaterial implantation [25,63]. Host derived chemoattractants released from activated platelets, endothelial cells, mast cells and injured cells direct neutrophils to the site of implantation. Neutrophils trigger a phagocytic response, degranulation and secrete ROS and proteolytic enzymes. The latter can corrode material surface and the degradation remnants could also trigger waves of neutrophils arrival and prolong the inflammatory response (reviewed in [63]. This event could also be occurring in our implanted PLGA membrane since it is susceptible to degradation when implanted in the body or in vitro [15,63,64]. Indeed, neutrophils were present in the foreign body reaction during all timelines of the study. In addition, the persistence of neutrophils in the reaction is a source of chemokines and activations factors for monocytes, macrophages, immature dendritic cells and lymphocytes [65,66]. Effectively a progressive influx of lymphocytes was verified in the late stages. Furthermore, activated lymphocytes at the implantation site could be the source of IL4 and IL13, both known to favor the macrophage fusion on biomaterials [67].

FBR are present within the implantation bed of different biomaterials including collagen-based biomaterials [68,69,70]. As already described, macrophage fusion is dependent on the presence of the fusogenic molecules on their surface but also on the environmental signals such as the quality and quantity of the adsorbed proteins in the provisional matrix on the biomaterial, on the surface itself and on the topography of the biomaterial surface [48,71]. Few reports elucidate the direct effects of the biomaterial surface on foreign body giant cells formation [72]. It was shown that decrease in monocyte adhesion and foreign body giant cell formation occurs in hydrophilic, anionic and nonionic polyacrylamide/polyacrylic acid surfaces compared to hydrophilic and hydrophobic, cationic surfaces [73]. Besides that, smooth flat surfaces induce considerably more foreign body giant cell formation than rough surfaces [27], while other reports relate that larger PLGA microspheres of 30 mm induce more foreign body giant cell formation than smaller 6 mm microspheres [74]. The implantation of porous materials constituted by high surface-to-volume is prone to show higher ratios of macrophages and foreign body giant cells than smooth-surface implants [75].

Albeit the observed long-standing foreign body response induced by the electrospun PLGA membranes, a fibrous capsule was never seen during all time points studied. Absence of fibrous encapsulation was previously found after the implantation of non-resorbable biomaterials [70] and was also reported in the subcutaneous tissue of rats implanted with PLGA [20,76,77]. Mitragotri and Lahann [75] reported that greater numbers of macrophages and foreign body giant cells on the surface of implants develop more fibrosis and encapsulation of the biomaterials. In addition, Whitaker [78] discussed that high levels of protein adsorption leads to increased cell adhesion and therefore, increased fibrous encapsulation. Considering the aforementioned concepts regarding fibrosis reaction, we can hypothesize that the 3D electrospun porous scaffolds used herein for implantation may have not had high levels of protein adsorption, and may have had decreased cell adhesion, resulting in no fibrous encapsulation. On the other hand, Al-Maawi [79] reported that the accumulation of macrophages and foreign body giant cells on the surface of a biomaterial is correlated with the formation of a highly vascularized granulation tissue and neovessels in the connective tissues. Moreover, Madden [80] reported that, nonporous scaffolds or those with 20 µm pores led to a significant increase in the fibrous capsule thickness, and pores of approximately 30–40 µm reduced the fibrous capsule thickness and the number of M1 macrophages. Several morphological variants of foreign body giant cells are already described but only the LCs and MNGCs participate on the foreign body response to biomaterials. The induction of granulomas around glass implanted in the dorsal subcutaneous tissue of rats allowed to demonstrate that LCs are precursors of MNGCs, and that the epithelioid cells derived from mature macrophages [62]. In view of these arguments, our study suggests that the tissue response induced by the electrospun PLGA membranes could reflect the constant arrival of monocytes to the implantation bed with LCs formation associated to the persistence of fusogenic molecules originated from activated macrophages without fibrous capsule formation.

In the recent years, researchers have become more focused on designing surface-modified PLGA membranes to minimize immune responses. Research studies from Kim et al. [20], Huang et al. [81] and Lee et al. [82], who have implanted PLGA 75:25; 50:50; 75:25, in rats, reported foreign body giant cells formation in intimate contact with PLGA, and decreased inflammatory cells in PLGA-coated membranes in comparison with PLGA alone. In the present study, foreign body giant cells were formed among the inflammatory cells over time. We hypothesized that the degradation products of PLGA 85:15 produced herein induced an inflammatory response in the site of implantation. 

While it is difficult to determine the specific role of any individual factor, it is important to highlight that surface topography, surface chemistry and surface energy may influence protein adsorption, platelet activation, cell growth and biocompatibility [83]. In particular, Adabi et al. reported that for some applications, hydrophobic surface could be preferred, since hydrophobic patterns showed more adsorption on intestinal mucus [83]. Since, our future challenges for the use of this membrane as a dressing material aim to accelerate the healing process of oral ulcerations in hamsters’ model, we chose a hydrophobic membrane for better adherence on the oral mucus of the animals. Another important issue refers to biocompatibility. Biocompatibility is not only polymer’s intrinsic property-dependent, but also biological environment-dependent [4]. For this reason, the intensity and length of specific polymer-tissue interactions can vary greatly in different organs, tissues and species [4,19]. The device used herein was implanted in hamsters to appreciate the immune responses due to our future objective of working with this animal model to develop new future dressing for chemotherapy-induced oral mucositis ulcerations. Indeed, hamster models have been extensively studied to observe pathological effects of radiation exposure or chemotherapy induced oral mucositis and help in the development of effective treatments [84,85,86,87]. Furthermore, Syrian golden hamsters seem to be an ideal animal model due to their low cost, small size, easiness to handle and ability to accurately reflect disease progression in humans [88]. Therefore, we used hamsters to verify the biocompatibility of the material. However, there remains a lack of available reagents for studying hamster immune responses which remains an issue to better characterize the type of immune response generated, critical for understanding protection from disease.

It is worthy to mention that, regarding our future challenges of applying the 3D electrospun PLGA (85:15) membrane with the addition of primary fibroblasts as a cell delivery dressing, we performed in vivo tests using a device with a higher surface area to volume which may lead to a higher degradation of the matrix [19]. 

Moreover, it is also important to mention that the accumulation of PLGA degradative products could cause significant host inflammatory response, a microenvironment favoring tissue fibrosis that is mainly mediated by M1 subtype macrophage [82]. However, considering our in vivo experiments in the present study, the scaffolds became well vascularized and there were no evidence of necrosis or encapsulation, which would have been strong contraindications for future clinical applications [72].

## 5. Conclusions

The present study appreciated cellular and tissue responses regarding PLGA (85:15) membranes produced by the 3D electrospinning method, for future applications in regenerative medicine. For this goal, both in vitro experiments were performed with primary fibroblast-like cells or MDCK cells that showed proliferation over PLGA electrospun membranes along time. In the in vivo experiment, cell-free irradiated PLGA membranes resulted in a chronic granulomatous inflammatory response in all time points after implantation, mainly constituted of epithelioid cells and LCs. Lymphocytes, myeloperoxidase**^+^** cells, LCs and capillaries decreased after 90 days post implantation. Light microscopy revealed foreign body giant cells showing internalized materials, without fibrous tissue formation. TEM analysis also showed cells exhibiting internalized PLGA fragments decreasing over time. Accordingly, we can conclude that MNGCs constituted a granulomatous reaction around the polymer, which resolved over time, probably preventing a fibrous capsule formation. We expect that the results of this work will boost other studies for translational applications of electrospun fiber membranes as cutaneous or oral dressings.

## Figures and Tables

**Figure 1 polymers-14-04460-f001:**
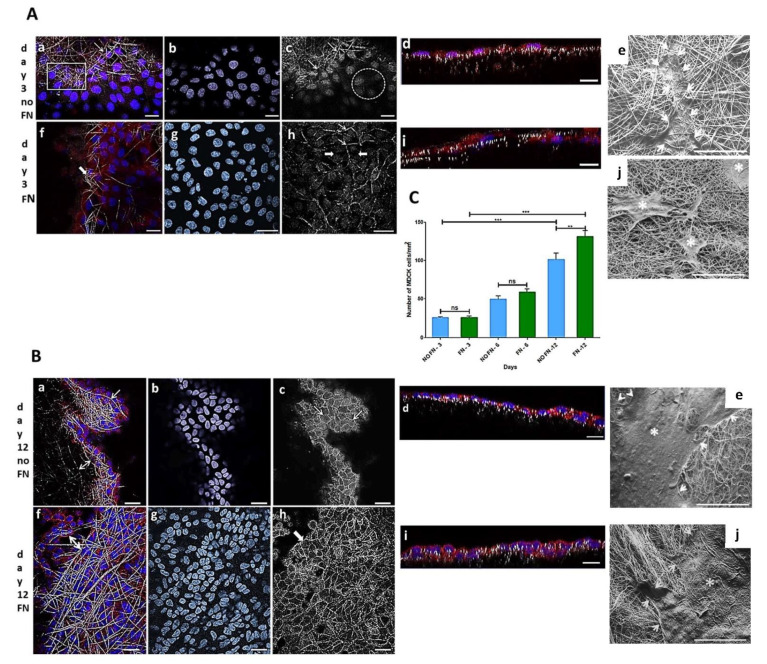
Representative images of MDCK cells cultured onto fibronectin-coated (FN) or uncoated (no-FN) PLGA membranes. (**A**,**B**): Confocal Laser Scanning Microscopy images of cells after 3 and 12 days, stained with Hoechst (blue) and phalloidin conjugated to Alexa 568 (red); (**C**): Graphical analysis. (**Aa**): Merge image of cells among no-FN PLGA fibers (arrow, square) after 3 days; (**Af**): Merge image of FN PLGA fibers and cells (white arrow) after 3 days. (**Ab**,**Ag**): Equivalent fields from (**Aa**,**Af**) processed on Image J software to visualize nuclei. Note the different organization of nuclei following the fibers after 3 days. (**Ac**): confocal images of phalloidin staining of no-FN coated fibers after 3 days (**Ac**, arrows). Note few circumferential actin belt staining revealing adhesions between cells (circle) in comparison with FN membranes (**Ah**) after 3 days showing more adhesions between cells (thin arrows), among cells without adhesions (thick arrows). (**Ad**,**Ai**): cells in a X-Z plane after 3 days forming a monolayer with less cells. Scale bars (**Aa**–**Ai**): 20 µm. (**Ae**,**Ai**): SEM images of no-FN or FN membranes exhibiting isolated cells onto the fibers after 3 days (white arrows and asterisks). Scale bars: 100 µm. (**Ba**,**Bf**): Merge image of cells after 12 days. Note fibers and cells among the fibers on no-FN or FN, respectively (white arrows and double arrows). (**Bb**,**Bg**): Equivalent fields from (**Ba**–**Bf**) showing a greater number of nuclei with a differential organization after 12 days. (**Bc**,**Bh**): Equivalent fields from (**Ba**–**Bf**) showing the increase in circumferential actin belts demonstrating formation of adhesions between cells exhibiting a cuboidal morphology onto PLGA fibers. Note a greater number of cells onto FN coated membrane after 12 days. (**Bd**,**Bi**): cells in an X-Z plane after 12 days showing more proliferation. Scale bars (**Ba**–**Bi**): 20 µm. (**Be**,**Bj**): SEM image showing cells covering the majority of the membrane after 12 days (white arrows and asterisks). Scale bars: 200 μm. (**C**): Graphical representation of number of MDCK cells/mm^2^ onto FN or no-FN PLGA membranes. ns: not significant, ** *p* < 0.001, *** *p* < 0.0001.

**Figure 2 polymers-14-04460-f002:**
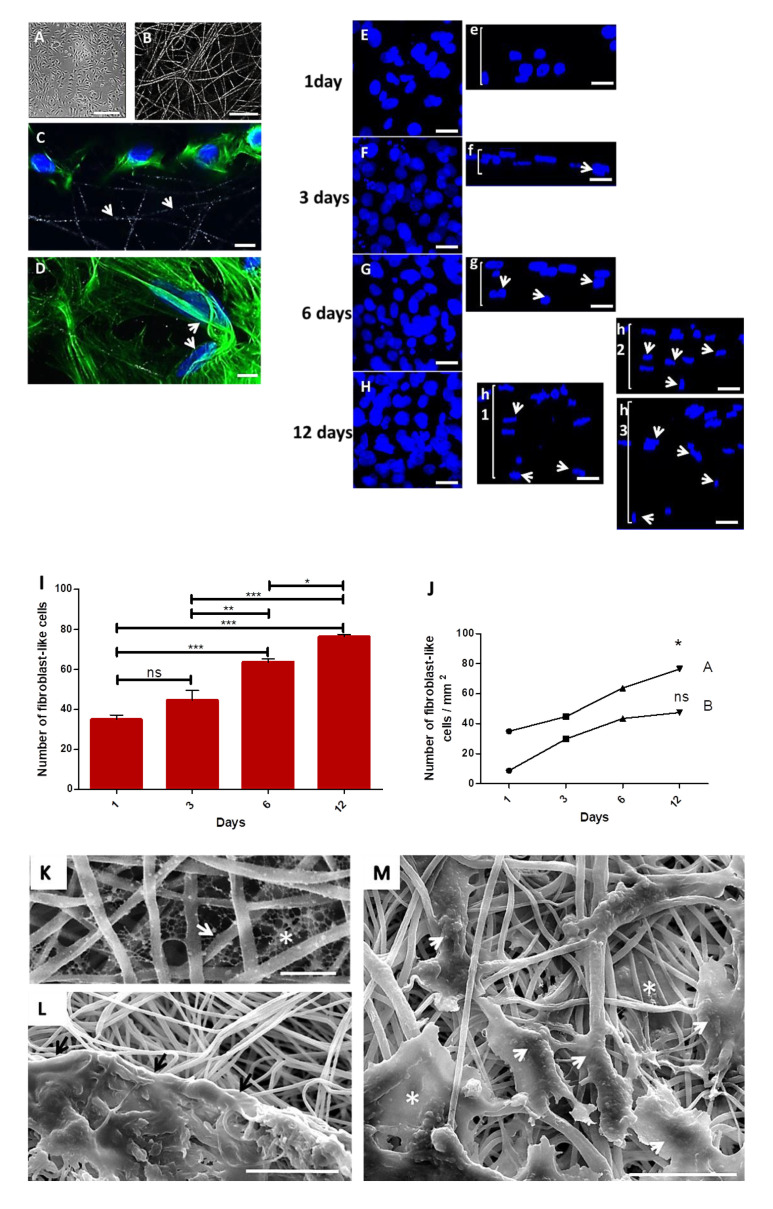
Primary hamster cheek pouch fibroblast-like cells cultured onto PLGA scaffolds coated with collagen I. (**A**): Phase-contrast image of primary fibroblast-like cells after 4 days. Scale bar: 100 μm; (**B**): CLSM image of collagen I-coated PLGA fibers. Scale bar: 20 μm; (**C**): CLSM image of fibroblast-like cells (3 × 10^5^) onto PLGA fibers after 1 day (white arrows). Nuclei in Hoechst stain (blue) and cytoskeleton in conjugated Alexa 488 phalloidin stain (green) for actin. Scale bar: 30 μm. (**D**): Cells adhered along different directions of PLGA fibers (white arrows). Scale bar: 30 μm. (**E**–**H**) CLSM images of fibroblast-like cells nuclei exhibiting proliferation after 1, 2, 3, 6 and 12 days on sample surface; (**Ee**,**Ff**,**Gg**,**Hh**): Side view of the nuclei of cells grown onto the scaffold after 1 day (**Ee**), and the in depth migration of cells nuclei to the 3D directions of the scaffolds after 3 days (**Ff**, arrow), 6 days (**Gg**, arrows) and 12 days (**Hh**, arrows)—images of 3 different fields in the same sample exhibiting a great amount of cells through the bulk of the membrane, as depicted in images h1,2,3 (arrow heads); note cells going deeper in image h3; Scale bars: 20 μm. (**I**): Graph of the proliferation of cells onto collagen I-coated membranes seeded with 3 × 10^5^ cells; (**J**): Graph of the expansion of cells in function of the quantity of inoculated cells: 10^4^ (A) or 3 × 10^5^ cells (B). *** *p* = 0.0001; ** *p* = 0.001 and * *p* = 0.01; ns: not significant; (**K**): SEM image of PLGA fibers (white arrow) showing the network of collagen (white asterisk) in the interfibrillar spaces (scale bar: 5 μm). (**L**): SEM image of cells in contact with PLGA fibers (black arrows). Scale bar: 30 μm. (**M**): SEM image of elongated cells following PLGA fibers (white arrowheads), and cells in interfibrillar spaces showing different morphologies (white asterisks) (scale bar: 30 μm).

**Figure 3 polymers-14-04460-f003:**
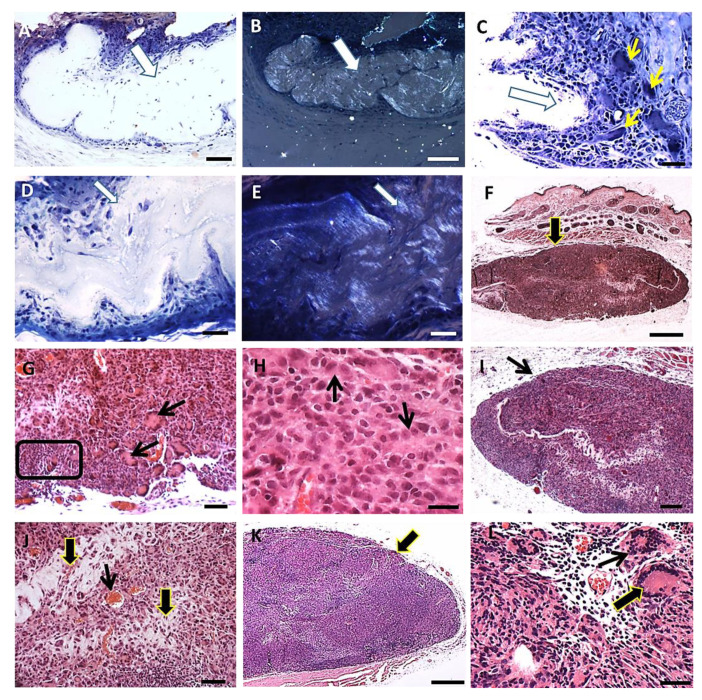
FBR induced by the PLGA membrane: histological features. (**A**): Semi-thin section of a sample in Spurr resin in toluidine blue stain showing the inflammatory response surrounding the membrane (thick arrow) 7 days post implantation (PI). Scale bar: 100 µm; (**B**): Polarization light microscopy of the same image in A. Scale bar: 200 µm; (**C**): Semi-thin section in Spurr resin and toluidine blue stain showing mononuclear cells and MNGCs (thin yellow arrows) surrounding the membrane (thick arrow) 7 days PI. Scale bar: 100 µm; (**D**): Semi-thin section in Spurr resin in toluidine blue stain showing the FBR 15 days PI. Scale bar: 100 µm; (**E**): Polarization light microscopy of the same image in D. Scale bar: 100 µm; (**F**): Section in Masson’s trichrome stain showing the FBR in the hypodermis 15 days PI (arrow). Scale bar: 1 mm; (**G**): Section in Masson’s trichrome stain showing the FBR 15 days PI; note LCs (arrows) and lymphocytes (square). Scale bar: 200 µm; (**H**): Section in HE stain showing epithelioid cells (arrow) surrounding the membrane. Scale bar: 50 µm; (**I**): Section in HE stain showing the FBR in the hypodermis after 30 days PI (arrow). Scale bar: 500 µm; (**J**): Section in HE stain showing the inflammatory response 30 days PI; connective tissue (thick arrows) and capillaries (thin arrow). Scale bar: 100 µm; (**K**): Section in Masson’s trichrome stains showing the FBR in the hypodermis after 90 days PI; note more cells (arrow). Scale bar: 1 mm; (**L**): Section in HE showing MNGC (thin arrow) and LC (thick arrow) in the inflammatory reaction 90 days PI. Scale bar: 50 µm.

**Figure 4 polymers-14-04460-f004:**
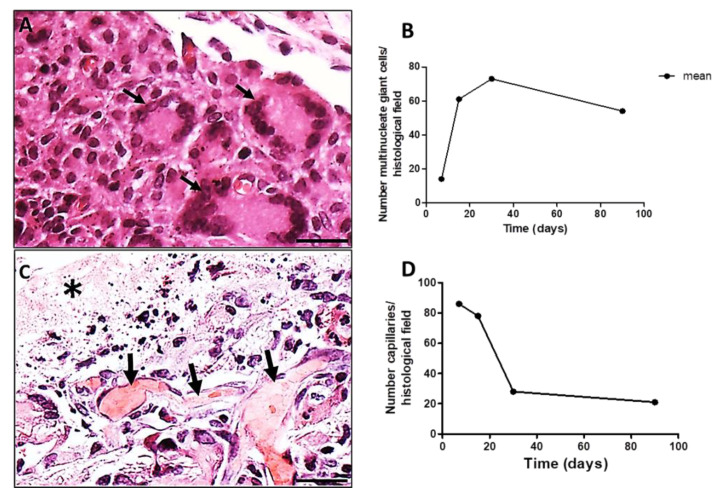
Histological sections of chronic inflammatory response induced by PLGA membranes. (**A**): Section showing the presence of LCs surrounding the chronic inflammation, 30 days PI. Scale bar: 50 µm; (**B**): Graphical representation of the number of LCs per histological field over time, showing decreased numbers up to 90 days PI. Number of MNGCs were calculated on 20 photomicrographs randomly obtained from HE stained sections (objective lens 20×). Results were expressed as number of MNGCs/histological field ± standard error of the mean; (**C**): Section showing the presence of capillaries (black arrows) along the evolution of the implanted PLGA (asterisk). Scale bar: 100 µm; (**D**): Graphical representation of the number of capillaries over time showing decreased number of capillaries up to 90 PI. Number of capillaries were calculated on 20 photomicrographs randomly obtained from HE stained sections (objective lens 20×).

**Figure 5 polymers-14-04460-f005:**
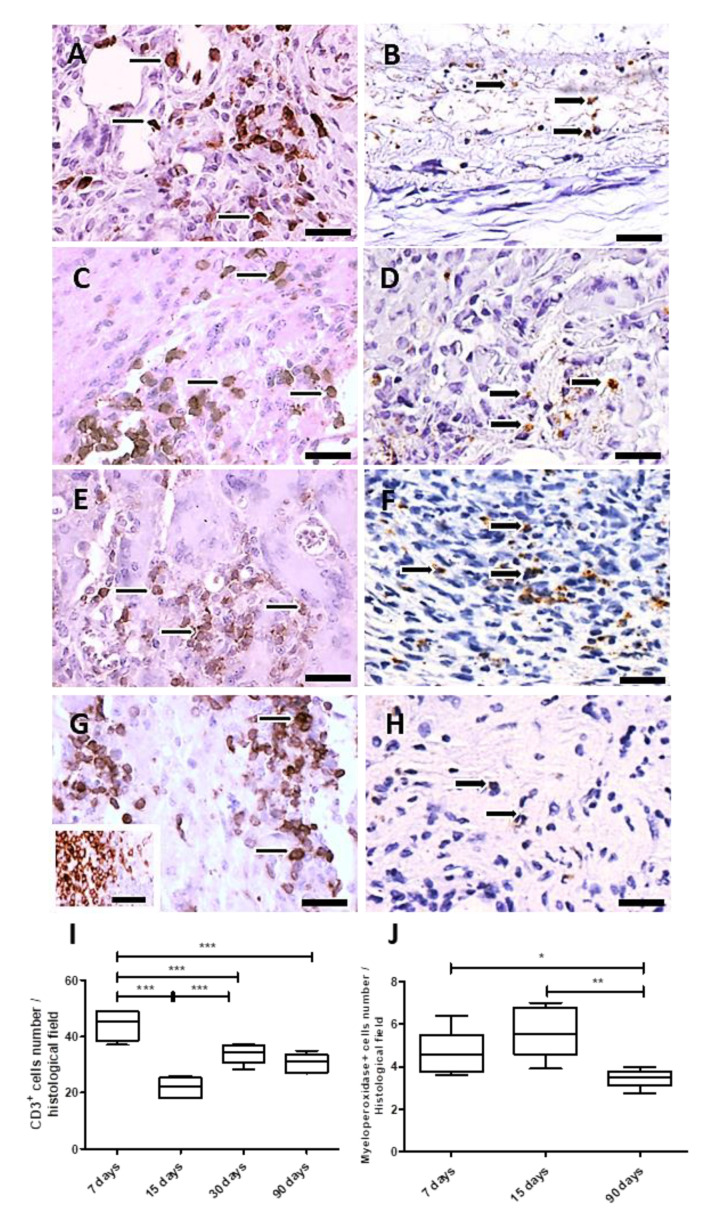
Immunohistochemistry and quantification of CD3 and myeloperoxidase positive cells. Detail of histological sections of PLGA-induced inflammatory response containing CD3^+^ cells. (**A**): 7 days PI (thin arrows); (**C**): 15 days PI (thin arrows); (**E**): 30 days PI (thin arrows); (**G**): 90 days PI (thin arrows). Detail of histological sections of PLGA-induced inflammatory response containing myeloperoxidase^+^ cells: (**B**): 7 days PI (thick arrows); (**D**): 15 days PI (thick arrows); (**F**): 30 days PI (thick arrows); (**H**): 90 days PI (thick arrows). In (**G**), inset represents the positive control of the anti-CD3 antibody, depicting CD3^+^ cells in human tonsil. Scale bars (**A**–**H**): 50 μm. (**I**): Graphical representation of the CD3^+^, showing decreased numbers at 15 and 90 days PI. (**J**): Graphical representation of myeloperoxidase^+^ cells along time showing increased number of myeloperoxidase+ cells a 15 days PI, decreasing 90 days PI. * *p* < 0.05, ** *p* < 0.01, *** *p* < 0.001, ns, indicates significant differences.

**Figure 6 polymers-14-04460-f006:**
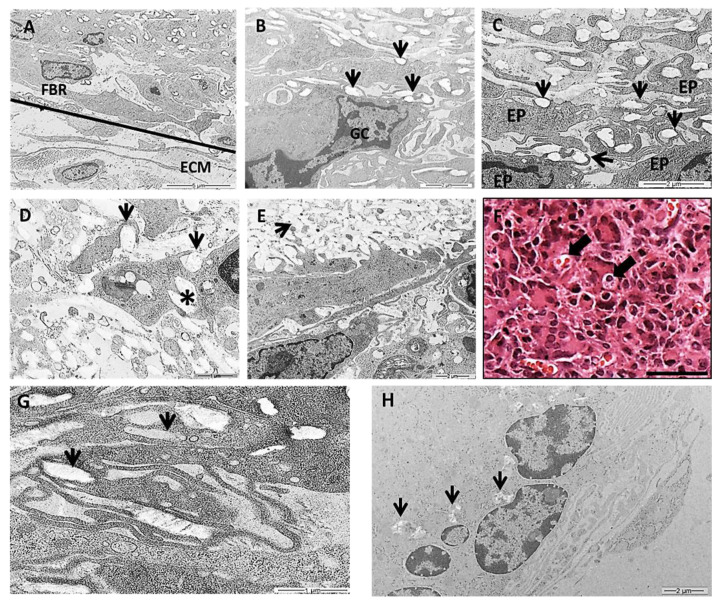
Transmission electron microscopy (TEM) and histological images of the inflammatory response induced by PLGA membranes. (**A**): Image showing the periphery of the PLGA with mononuclear cells forming a FBR in contact with the extracellular matrix (ECM). Scale bar: 5 μm; (**B**): image showing PLGA fragments (arrows) in contact with a MNGC. Scale bar: 2 μm; (**C**): image showing PLGA fragments (arrows) in contact with EP cells taking different morphologies, exhibiting prominent cytoplasmic extensions such as filopodia following the topography of PLGA fragments (arrows). Scale bar: 2 μm; (**D**,**E**): 15 days PI. D: PLGA fragments inside phagosomes (arrows in **D**) in the interior of the cell (asterisk in **D**) (Scale bar: 1 μm) or in the extracellular space (arrow in **E**). Scale bar: 2 μm; (**F**): Light microscopy image 30 days PI in Masson’s trichrome staining showing LC containing material inside (arrows). Scale bar: 100 µm; (**G**): 30 days PI—PLGA fragments were swollen by cells (arrows). Scale bar 1 μm. (**H**): Note less amount of PLGA fragments between the cells 90 days PI (arrows). Scale bar: 2 μm.

**Table 1 polymers-14-04460-t001:** Characteristics of antibodies used and antigenic recovery in the immunohistochemistry assays.

Antibody	Manufacturer	Antigenic Recovery	Dilution
CD3	Dako, polyclonal rabbit, cat.# GA503	Steamer 20 min, Citrate Buffer 0.01 M pH 6.0	1: 400
Myeloperoxidase	AbCam, CA, USA, polyclonal rabbit, cat. # ab9535	Microwave—3 min 3 x, Citrate Buffer 0.01 M pH 6.0	1:100

## Data Availability

The raw/processed data required to reproduce these findings can be shared upon request.

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
