# Peer review of "In Vitro and In Vivo Cell-Interactions with Electrospun Poly (Lactic-Co-Glycolic Acid) (PLGA): Morphological and Immune Response Analysis"

_polymers, 2022, doi:10.3390/polym14204460_

Round 1
Reviewer 1 Report
The paper "In vitro and in vivo Cell-Interactions with Electrospun Poly2 (Lactic-Co-Glycolic Acid) (PLGA): Morphological and Immune Response Analysis" is highly interesting for scientist working in the area of tissue engineering and with biodegradable polymers.
However some issues need to resolved in order to ameliorate the quality and significance of this paper.
In details
The authors use the Journal form, but font and line spacing are neither respected, nor uniform along the whole manuscript. Please correct.
The authors used PLGA 85:15. Please note that PLGA exhists in various compositions % of PLA and PGA, and the immune reaction can change depending on the polymer composition. Therefore I suggest to authors to specify that their results cannot generally addressed to PLGA, but only to PLGA 85:15.
The author should better explain which is the rationale of using MDCK cells.
All Figure legends are too much long. They need to be simplifies and shortened and results should be reported as results, not only as figure legend.
Author Response
Dear Reviewer,
Please find enclosed the final version of the manuscript, the answers, and the new format of the graphical abstract.
Best regards,
Ana Chor

Reviewer 2 Report
Comments
To authors
In this work, the authors fabricated the PLGA fibrous membrane coated with fibronectin or collagen and study the cell behavior and immune response in vivo. But the content of manuscript is relatively bland. The group of pictures and narration are confusing which were need to be major revised.
1. Lack of relevant literature support.
Page 2, line 73-74, ‘Recently it was shown that, when covered with nanoparticles or bioactive factors, PLGA membranes can turn into a promising bifunctional scaffold’
Page 2, line 74-75, ‘…..those scaffolds play a crucial role in cell migration and differentiation……’
2. Even though the material has good cellular adhesion behavior, severe immune response is not conducive to inducing normal tissue function. Tissue repair and collagen growth in different period of time (7, 15, 30, 90d) need further evaluation.
3. At present, there are many researches on regulating cell behavior or inflammatory response by controlling the structure/surface modification of biomaterials. What is the difference/novelty based on this manuscript when comparing with other published articles.
4. What was the main factor affecting the behavior of the cells in the study? PLGA fiber structure or collagen coating on the surface?
5. The evaluation part basically stays at the phenomenon and lacks explanation of mechanism. It would be more convincing to do further mechanism explanation and verification.
6. The typography of the figure is very confusing, and the text on the specific pictures is lack or too small, which reduces the readability of this article. For example
Figure 1: The results of different groups (with or without FN) should be marked with text and arranged vertically (or horizontally) so as to provide an intuitive comparison.
Figure 2, 3: It is important to use text to explain the group directly in figure.
7. The format of figure legend is not unified, please modify it according to the journal requirements.
8. The use of punctuation needs to be improved. In some places, the incorrect use of symbols and the lack of capital letters caused confusion and reduced readability. Please check the full text and correct it. For example,
Page 7, line 338, ‘towards areas without cells (Figure 1-L).’ There is no Figure 1-L.
Author Response
Dear reviewer,
Please find enclosed the answers, the final version of the manuscript, and the new format of the graphical abstract.
Best regards,
Ana Chor

Reviewer 3 Report
This manuscript comprehensively studied the in vitro and in vivo cell interactions of electrospun PLGA membranes after host implantation. The author has applied morphological, cytochemical, and immunohistochemical approaches associated with laser and electron microscopy. Collectively they looked into cell proliferation over time, and tissue immune responses after membrane implantations. The manuscript is thorough, well presented, and has a study with good analysis and reasonable conclusions. I recommend this study for publication in Polymers after some minor corrections.
In the introduction, the author should discuss PLGA-based implants currently area of interest or under clinical investigation.
Figure 4B, 4D missing standard error bar.
Is the immune response going to be different if the implant is at a different position in the body?
In conclusion, the author needs to discuss how their findings are going to help other researchers.
Author Response

(The authors gave the same response as above.)

Round 2
Reviewer 2 Report
1. Text paragraphs and images can be aligned with the context, uniformly centered or to the side to increase readability
2. The Chapter 5 named Conclusion can be appropriately segmented with two indented Spaces at the beginning to facilitate readers' understanding
3. the contrast of the image can be adjusted to make the structure highlighted by the arrow clearer
Author Response
Response to Reviewer 2 Comments
Point 1 - Text paragraphs and images can be aligned with the context, uniformly centered or to the side to increase readability.
Answer 1: Text paragraphs and images were aligned uniformly centered, except
figures 4 and 5 because they move through the text.
Point 2: The Chapter 5 named Conclusion can be appropriately segmented with two indented Spaces at the beginning to facilitate readers' understanding
Answer 2: It was corrected.
Point 3: the contrast of the image can be adjusted to make the structure highlighted by the arrow clearer
Answer 3: The contrast was improved in all figures.
